



# Nature and origin of variations in pelagic carbonate production in the tropical ocean since the Mid Miocene (ODP Site 927)

Pauline Cornuault[1], Thomas Westerhold[1], Heiko Pälike[1], Torsten Bickert[1], Karl-Heinz Baumann[2], Michal Kucera[1]

[1]University of Bremen, MARUM - Centre for Marine Environmental Sciences, Leobener Straße 8, D-28359 Bremen, Germany

[2]University of Bremen, Geoscience Department, Klagenfurter Straße, PO Box 330440, 28359 Bremen, Germany

*Correspondence to: pcornuault@marum.de*

**Abstract**

Marine plankton is an important component of the global carbon cycle. Whereas the production and seafloor export of organic
carbon produced by the plankton, the biological pump, has received much attention, the long-term variability in plankton calcification, controlling the carbonate counter pump, remains less well understood. Yet, it has been shown that on geological time scales, changes in pelagic calcification (biological compensation) could affect the ocean's buffering capacity and thus regulate global carbon budget. Here we use Neogene pelagic sediments deposited on the Ceara Rise in the tropical Atlantic to characterise the variability in pelagic carbonate production with focus on warm climates. A re-evaluation of published records
of carbonate accumulation at Ceara Rise reveals a systematic increase in sedimentation rates since the late Miocene, but the carbonate accumulation rate does not show a clear trend. Instead, we observe substantial orbital time-scale variability in carbonate accumulation and an effect of carbonate preservation, especially at sites located below 4 km. To evaluate long-term changes against possible orbital-scale variability, we generated new high-resolution records of carbonate accumulation rate at ODP Site 927 across two Quaternary interglacials (MIS 5 and MIS 9), the Pliocene warm period (MIS KM5) and the Miocene
climate optimum (MCO). We observe that the highest carbonate accumulation rates occurred during the Pliocene but that each interval was characterised by large orbital-scale variability. Prominent variations in carbonate accumulation prior to the Quaternary preservation cycles appear to follow obliquity and eccentricity. These results imply that pelagic carbonate production in the tropical ocean, buffered from large temperature changes, varied on orbital time scales similarly or even more than on longer time scales. Since preservation can be excluded as a driver of these changes prior to the Quaternary, the observed
variations must reflect changes in the export flux of pelagic biogenic carbonate. We conclude that the overall carbonate production by pelagic calcifiers responded to local changes in light, temperature and nutrients delivered by upwelling, which followed long orbital cycles, as well as to long-term shifts in climate and/or ocean chemistry. The inferred changes on both time scales were sufficiently large such that when extrapolated on a global scale, they could have played a role in the regulation of the carbon cycle and global climate evolution during the transition from the Miocene warm climates into the Quaternary
icehouse.

**Keywords :** Pelagic production, Carbonate, Carbon cycle, Alkalinity, Warm conditions

## 1 Introduction

The ocean plays a key role in the climate system as one of the major sinks for anthropogenic atmospheric $CO_2$ (Landschützer et al., 2014). Most of the excess atmospheric carbon ($CO_2$) is absorbed by the ocean as dissolved $CO_2$ which becomes part of
the seawater carbonate system and can be sequestered by the metabolic activity of marine organisms. Large part of the carbon



sequestration is due to carbon fixation into organic matter by photosynthesis (Henson et al., 2012; Passow and Carlson, 2012; Sarmiento et al., 2004). However, next to the sequestration of $CO_2$ by photosynthesis and export via the biological pump, marine organisms also participate in the global carbon cycle by carbonate biomineralisation. Milliman (1993) estimated that today's marine carbonate production by organisms amounts to 5.3 GT $yr^{-1}$ of which about a half is accounted for by pelagic

calcifiers (2.4 GT $yr^{-1}$). Since aragonite and high Mg calcite are unstable and largely dissolve before deposition, the geologically relevant aspect of the pelagic biogenic carbonate production is mediated mainly by low Mg calcite, that may be variable although mostly dominated by both planktic foraminifera and coccolithophores (Boudreau et al., 2018). The carbonate biomineralisation, also termed carbonate counter-pump, leads on the short term to the release of $CO_2$ from seawater, because it consumes alkalinity, but on long, geological time scales, it sequesters carbon from the dissolved volatile ocean-atmosphere

reservoir into the more inert sedimentary reservoir. Manipulative experiments, ocean chemistry profiles and numerical models all indicate that pelagic carbonate production is affected by a range of environmental parameters, such as temperature, nutrient availability or $pCO_2$ (Feely, 2004; Gehlen et al., 2007). Therefore, a change in any of these parameters could impact the pelagic carbonate production resulting in a process that Boudreau et al. (2018) termed biological compensation. In contrast to chemical compensation, where changes in ocean carbonate chemistry are compensated by dissolution of seafloor carbonate deposits,

biological compensation refers to changes in ocean carbonate chemistry due to globally relevant shifts in carbonate biomineralisation. For example, a decrease in global oceanic biomineralisation would lead to an increase of alkalinity, which would cause a increase $CO_2$ solubility and therefore lead to an increased capacity of the ocean to take up $CO_2$ (Boudreau et al., 2018; Sarmiento and Gruber, 2006). Using a modelling approach, Boudreau et al. (2018) showed that a global carbonate productivity change by only 10 % would be sufficient for the process of biological compensation to affect the marine carbon

cycle on time scales from years to millions of years.

For the process of biological compensation to play an important role in the global carbon cycle, it must be demonstrated that sufficiently large changes in global carbonate biomineralisation occurred in the geological past. However, measuring changes in global biogenic carbonate production is difficult, because productivity and biomineralisation vary in space, and changes observed in individual records could be compensated by complementary shifts elsewhere in the ocean (Drury et al., 2020). In

most parts of the ocean, climate change causes plankton assemblages to migrate, with biogeographic provinces expanding and contracting in pace with orbital cycles (Yasuhara et al., 2020). These processes should result mainly in the spatial reorganisation of pelagic carbonate production and as long as the forcing is cyclic, the effects should cancel out over time.

Beyond orbital time scales, understanding of changes in carbonate production are complicated by the confounding effects of biological and chemical compensation on carbonate content of deep-sea sediments (Boudreau et al., 2018). Nevertheless, the

few existing continuous records indicate the presence of long-term shifts in carbonate production by a factor of two or more manifested for example as the late Miocene carbonate maximum (Lyle et al., 2019; Drury et al., 2020; Liebrand et al., 2016). Although there is abundant evidence for local changes in pelagic calcification and carbonate production, their spatial extent remains unknown, making it difficult to judge whether the local shifts may have resulted in globally significant biogeochemical response (Lyle et al., 2019; Drury et al., 2020).

Here we have investigated pelagic carbonate accumulation, as a proxy for production, in an equatorial location, where the plankton had no opportunity to respond to the climate cycles by migration and where long-term changes in temperature, a key parameter likely affecting biomineralisation, were muted (Herbert et al., 2016). Since orbitally driven environmental change still affected the tropics, the Cenozoic tropical plankton represents a natural experiment where the tropical calcifying community responded to a number of orbital cycles and long-term changes in ocean chemistry, reflecting changing atmospheric

$CO_2$. Whilst these records cannot provide a direct answer on how much pelagic carbonate production changed globally, they can provide a first-order constraint on the amount of change in pelagic calcification that could occur due to changes in the constitution and/or abundance of the calcifiers on different time scales. We specifically decided to target intervals with warmer global climate states, providing potential analogues to gauge the amount of change in tropical pelagic carbonate production



under a global warming scenario (Fig. 2), and the tropical Atlantic location allows us to complement records from the Pacific
and South Atlantic (Lyle et al., 2019; Drury et al., 2020; Pälike et al., 2006a) to assess the spatial coherence of long terms
trends and the amount and nature of short-term variability.

Next to analysing long-term changes in carbonate accumulation, the existence of persistent orbital variability implies that new
data will be required, characterising the short-term response of the tropical pelagic carbonate production system. To this end,
in the present study the changes of carbonate production through time have been studied in four warm periods of the late
Cenozoic: the marine isotopic stage (MIS) 5, the MIS 9, the MIS KM5 and the Miocene Climatic Optimum (MCO). This
approach allows us to evaluate long-term changes in pelagic carbonate production since the Mid-Miocene and at the same time
to characterise the orbital-scale variability, and determine if the orbital periodicity forcing carbonate production changed from
the Miocene to present.

### 1.1 Time intervals

The MIS 5 (Eemien), as the last warmest and longest interglacial of the past 500 ka (Howard, 1997), with an abrupt glacial-
interglacial transition (Howard, 1997; Müller and Kukla, 2004; Sirocko et al., 2005) is considered to be a good analogue for
the actual warm Holocene (Howard, 1997; Kukla, 1997) and even a partial analogue for + 1 - 2°C scenarios because of polar
temperatures 3 to 5°C warmer than today and a sea level about 6.6 m higher than today (Clark and Huybers, 2009; Kopp et
al., 2009). During this interglacial, Chalk et al. (2019) observed a change in the current circulation in the Atlantic Ocean, with
an enhanced Antarctic Bottom Water (AABW) below 3400 mbsl and well ventilated, high pH and [$CO_3^{2-}$] water mass around
2200 mbsl. They also highlighted a correlation between the [$CO_3^{2-}$] and the $pCO_2$ in the west Atlantic during cold intervals,
with an increase of the volume of the high dissolved inorganic carbon (DIC), low [$CO_3^{2-}$] deep water masses in the North
Atlantic.

The MIS 9 in the Equatorial Atlantic is presenting well-preserved sediment at a period known to be under high obliquity with
100   a unique insolation signal. Stable oxygen isotope values are low during this period (low ice volume). It is one of the interglacials
showing the highest $pCO_2$ (around 300 ppm) and $pCH_4$ (around 25 ppb) conditions in the late Pleistocene. This period is also
one of the warmest, stablest and shortest interglacials, with a weak surface water ventilation (Past Interglacials Working Group
of PAGES, 2016; Marino et al., 2014; Voelker et al., 2010).

The Pliocene warm period (PWP) MIS KM5 corresponds to a period with a similar orbital forcing to present day and an
105   insolation distribution close to the modern one (Haywood et al., 2013). This interval (3.264 ka - 3.025 ka) is also described as
a negative oxygen isotope slope and a 21 - 23 m sea-level above the present day one (Lunt et al., 2010, 2008; Naish et al.,
2009; Pollard and DeConto, 2009) with a deep Atlantic Ocean well ventilated (Bell et al., 2015). The temperature 3°C higher
than pre-industrial values (Haywood et al., 2000; Lunt et al., 2010) and the $CO_2$ concentration close to the present one – 330 -
425 ppmv during the warm interglacials (Pagani et al., 2010; Seki et al., 2010) – makes it a good analogue for future climate
(Ravelo and Wara, 2004) and an important period to understand the climate system (Lunt et al., 2010). Furthermore, this period
is also characterised as wetter (Leroy and Dupont, 1994; Dodson and Macphail, 2004) but the latitudinal distribution of the
rainforest was close to the present day one (Salzmann et al., 2011).

The MCO corresponds to a period with an eccentricity-modulated precession $\delta^{18}O$ signal and low global ice volume, with the
Northern hemisphere free of continental ice-sheets and important 100 and 400 ka orbital variability, and an Antarctic ice-sheet
smaller but more dynamic than today (De Vleeschouwer et al., 2017; Holbourn et al., 2007). Haq et al. (1987) highlighted a
large sea-level amplitude from 16 to 14 Ma and the annual global temperature was 3 to 8°C higher than today (Pound et al.,
2012; You et al., 2009). The climate during the MCO is known to be correlated with atmospheric $CO_2$ concentration changes
(Foster et al., 2012), with $CO_2$ concentration generally lower than at present (Foster et al., 2012; Ruddiman, 2010; Zachos et
al., 2008; Zachos, 2001b, a), but peaking at 16 Ma between 460 and 564 ppmv (Kürschner et al., 2008). Even if a decline in
$\delta^{18}O$ and $\delta^{13}C$ at 16.9 Ma was suspected to be linked to increase of carbonate dissolution, a sign of strong changes in the carbon





cycle (Holbourn et al., 2015), carbonate production appears to have been the main control of the CaCO$_3$ record (Liebrand et al., 2016).

## 2 Material and methods

### 2.1 Site location

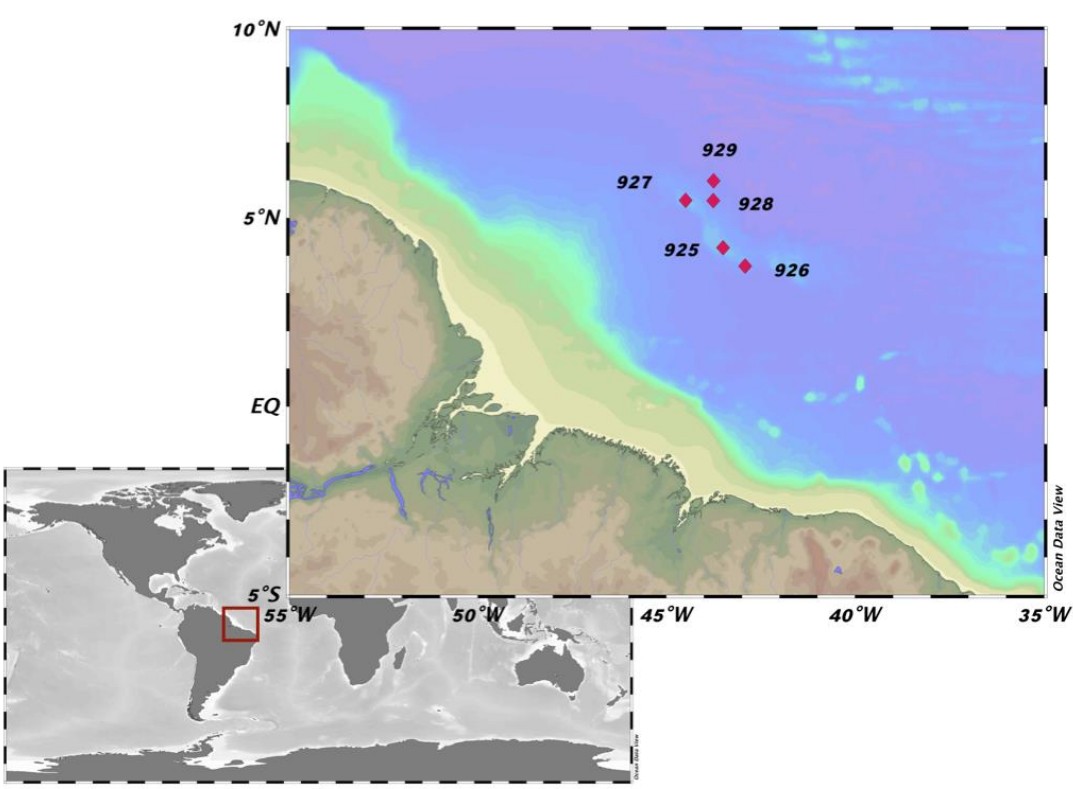


**Figure 1.** Location of the material of this study Ceara Rise, ODP Leg 154 (Ocean Data View V 5.1.7, Schlitzer et al., 2015).

Ceara Rise, located in the equatorial Atlantic Ocean, represents an ideal location to quantify the variability in tropical Atlantic
pelagic carbonate production since the Miocene. This aseismic ridge rises several km above the surrounding abyssal plain,
well above the regional lysocline (Frenz et al., 2006; Gröger et al., 2003a, b; Curry et al., 1995; Cullen and Curry, 1997)
providing an opportunity to sample pelagic sediments that are largely unaffected by dissolution and their accumulation
therefore mainly reflects changes in pelagic carbonate production as suggested by Brummer and van Eijden, (1992). The Ceara
Rise (Fig.1) has been visited by Ocean drilling program (ODP) Leg 154 (Curry et al., 1995), recovering a transect of sediment
sequences ranging into the Eocene, which are rich in carbonate and show prominent cycles due to variable input of clastic
material from the Amazon fan (Shackleton et al., 1999; Bickert et al., 1997; Shackleton and Crowhurst, 1997). The cycles are
reflected in sediment physical properties, such as colour or magnetic susceptibility, and because of the very good recovery and
repeated coring at the same sites, continuous spliced records could be produced that facilitated the development of orbitally
tuned age models (Shackleton et al., 1999; Zeeden et al., 2013; Wilkens et al., 2017; Shackleton and Crowhurst, 1997), a
prerequisite for the quantification of carbonate accumulation. Since all high-resolution Neogene records of carbonate



accumulation (Drury et al., 2020; Lyle et al., 2019), including those from the Ceara Rise (Curry et al., 1995; King et al., 1997) show a large orbital-scale variability, hinting at prominent orbital-scale variability in pelagic carbonate production, next to a compilation and re-evaluation of existing carbonate records, the selected time slices had to be newly sampled and analysed at higher resolution.

**2.2 Compilation of existing carbonate data from ODP Leg 154**

The combination of the availability of high-resolution age models and good carbonate preservation make the Ceara Rise a model region to study pelagic carbonate production and preservation. We compiled existing data on carbonate content ($CaCO_3$ %) at all the Leg 154 sites since the Miocene (Curry et al., 1995; Frenz et al., 2006; King et al., 1997) and used those to calculate carbonate accumulation rates (AR). The few other existing datasets on carbonate content from the Ceara Rise sites

(e.g., Tiedemann and Franz, 1997) could not be used because some of the information needed to calculate accumulation rates or the original samples ID and depths was not available.

The carbonate content data were combined with dry bulk density (DBD) and sedimentation rate (SR) to calculate the carbonate accumulation rate ($CaCO_3$ AR) as Eq. (1).

(1) $CaCO_3$ AR = ($CaCO_3$ % / 100) x DBD x SR

Following the approach by Lyle (2003), we first derived for each site a calibration between the gamma-ray attenuation (GRA) bulk density and DBD using data from Curry et al. (1995). The resulting site-specific calibrations (Fig. S1) were then applied on GRA bulk density values, which were extracted from Curry et al. (1995), and interpolated to the position of the analysed samples using linear interpolation. This yielded DBD values between 0.40 g cm$^{-3}$ and 1.64 g cm$^{-3}$. For two samples, the calibration returned negative DBD (at 129.62 mcd and 135.47 mcd) due to two anomalous GRA values below 1. In these cases,

we used the DBD of the nearest point instead.

**2.3 Context and sampling plan**

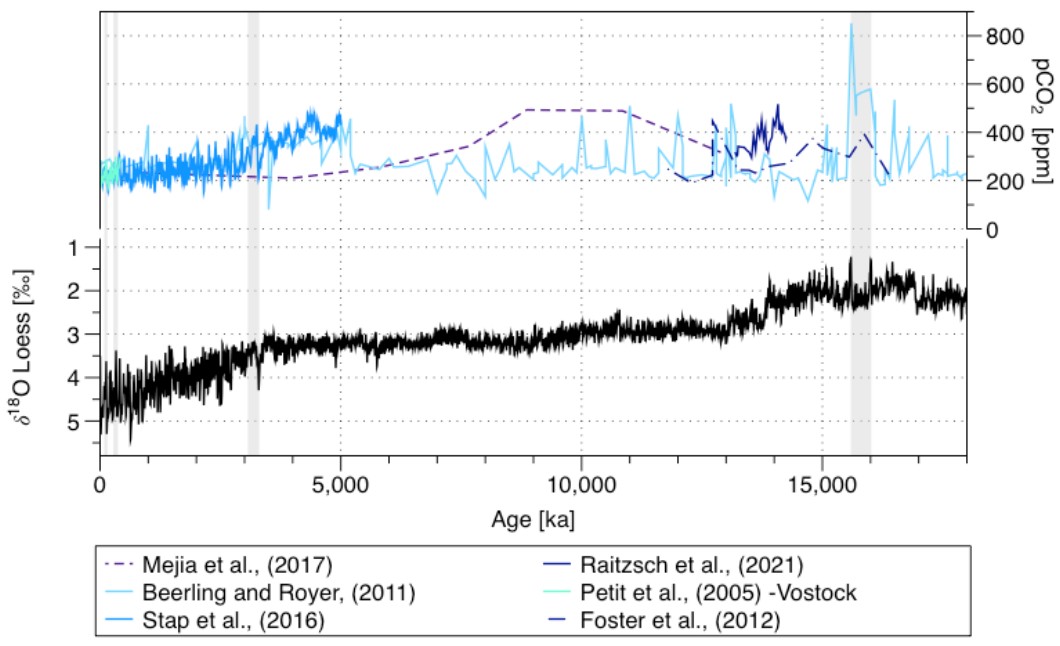



**Figure 2.** Stable oxygen isotopes ($\delta^{18}$O) (Westerhold et al., 2020) and pCO$_2$ (Mejía et al., 2017; Beerling and Royer, 2011;
Stap et al., 2016; Raitzsch et al., 2021; Foster et al., 2012; Petit et al., 1999) over the last 16 Ma and sampled intervals
(shadows).

Next, we sampled the record at Site 927 at high resolution for the four periods of interest (Fig. 2), making sure that for each
interval both the interglacial and the flanking glacial in the Quaternary and at least two full eccentricity cycles during the
Pliocene and Miocene have been covered. These four intervals are covering a large range of global temperature and CO$_2$ values
(Fig. 2). We selected Site 927 because it is one of the two shallow sites of Leg 154, located well above the lysocline at present
(Frenz et al., 2005; Curry et al., 1995; Bickert et al., 1997), and because numerous paleoceanographic datasets and carbonate
measurements exist for this site (e.g. Bickert et al., 1997; Pälike et al., 2006; Frenz et al., 2006; Gröger et al., 2003b; King et
al., 1997; Curry and Cullen, 1997). The site appears generally less affected by slumps or turbidites than the four others, which
were not observed in the four studied intervals (sampled out of the slumps and turbidites reported lithostratigraphic units)
(Curry et al., 1995). The sampling was guided by the Wilkens et al. (2017) age model for the samples from 0 to 14 Ma, and by
the Shackleton et al. (1999) age model of the samples from 14 to 16.5 Ma. Considering the typical mixing depth of 5 - 10 cm
in deep sea sediments, we sampled at 5 cm in the Quaternary and 10 cm in the Neogene, which in both cases provides sub-
orbital resolution. The resolution was higher in the Quaternary, because the peak interglacial warmth periods are short (<10
ka for MIS 5e; Stolz and Baumann, 2010; Müller and Kukla, 2004; Sirocko et al., 2005) and we wanted to cover these by
multiple samples. In total, we collected and analysed 139 samples for the two Quaternary intervals, 72 samples for the Pliocene
and 50 samples for the Miocene.

## 2.4 Stable isotopes analyses

We performed stable isotopes analyses ($\delta^{18}$O and $\delta^{13}$C) at Bremen university, using a ThermoFisher Scientific MAT 253plus
gas isotope ratio mass spectrometer with Kiel IV automated carbonate preparation device. This gives $\delta^{18}$O values with a
standard deviation of house standard (Solnhofen limestone) over measurement period of 0.07 ‰ and $\delta^{13}$C values with standard
deviation of house standard (Solnhofen limestone) over measurement period of 0.03 ‰. The sediment samples were washed
and sieved at 63 µm using tap water and dried overnight in the oven at 50°C. Then, they have been dry-sieved at 150 µm for
benthic foraminifera picking. All the Miocene samples have been picked, only 3 samples did not have enough material to run
the stable isotopes analyses. For some of the samples we had enough material to analyse two or three replicates using different
species known to be relevant markers for $\delta^{18}$O seawater: *Cibicidoides mundulus*, *Cibicidoides wuellerstorfi* and *Oridorsalis
umbonatus* (Katz et al., 2003; Rathmann and Kuhnert, 2008). We did not mix the species in one single measurement. For the
species-specific $\delta^{18}$O and $\delta^{13}$C values correction, we used the calibration given in the supplement table S3 from Westerhold et
al. (2020).

## 2.5 Age model

### 2.5.1 For the existing data compilation

Because the orbitally tuned age models as well as the splices for the individual sites have been recently revised (Wilkens et
al., 2017), we re-evaluated the composite depth of all samples and assigned new ages to them based on Wilkens et al. (2017)
and used the new ages to derive sedimentation rates (SR).

### 2.5.2 For the high resolution 4 intervals of core 927



The existing most recent age model for Site 927 is based on a directly tuned age model from Site 926 that has been point-to-point correlated with the composite record from Site 927, using core images, magnetic susceptibility, grey scale values and stable isotopes (Wilkens et al., 2017; Zeeden et al., 2013). For the determination of carbonate AR during the four target intervals, this age model requires adjustments, because it provides too low resolution and it is not tuned below core 927A-30H,

section 6, 70 cm (303.60 rmcd), corresponding to 926A-28H, section 3, 18 cm (277.82 rmcd). Thus, to estimate carbonate AR for the three studied intervals, we developed modified age models, where SR have been constrained directly by astronomical tuning of sediment properties in the studied cores.

**2.6 Carbonate analyses**

To determine the carbonate accumulation rates for the newly sampled intervals, we performed carbonate content analyses on

the bulk sediment using a LECO CS744 elemental analyser at Bremen University. The analysis was performed by heating 0.1 g of homogenised material in a ceramic dish and measuring the resulting $CO_2$ in IR cells. The carbonate content has been calculated as the difference between the total carbon content and the organic carbon content, measured in a second sample that was pre-treated with hydrochloric acid to remove carbonates. Both measurements have an accuracy of 0.001 mg (1 ppm) or 0.5 % relative standard deviation (RSD). The inorganic carbon was then converted to carbonate content using the molecular

mass of calcium carbonate. Dry bulk density for all the newly analysed samples at Site 927 was determined from GRA bulk density as described above (Sect. 2.2.) and combined with the carbonate content and SR from the modified age models to calculate the carbonate accumulation rates.

**3 Results**

**3.1 Long-term trends in carbonate accumulation rates**






**Figure 3.** a) CaCO$_3$ mean accumulation rates for the five cores and b) CaCO$_3$ accumulation rates and sedimentation rates (grey line) through the time for the Sites 925, 926, 927, 928 and 929 (black line and dots) for the five cores of the Leg 154. The CaCO$_3$ accumulation rates are calculated from existing carbonate content data for all Leg 154 sites (Curry et al., 1995; Frenz et al., 2006; King et al., 1997). The colour shade corresponds to the values of CaCO$_3$ AR.

Using existing carbonate content data for all Leg 154 sites (Curry et al., 1995; Frenz et al., 2006; King et al., 1997), combined with new age models (Wilkens et al., 2017), for each site, records of carbonate AR since the mid Miocene were calculated (Fig. 3). Curry et al. (1995) noted the occasional presence of slumps or hiatuses in the sediment sequences, especially at Site 928 and Site 929. Here we used the age models for the entire sediment package, ignoring the presence of these events. This is because the slumps only represent a small fraction of the sediment sequence and therefore are unlikely to affect the overall trends.



The mean CaCO₃ accumulation rate varies considerably among the sites, reflecting their depth and therefore likely the amount of dissolution. ODP Sites 925 and 927 (present depth 3041 mbsl and 3315 mbsl) show consistently higher CaCO₃ AR (between

1.5 and 3 g cm$^{-2}$ ka$^{-1}$) than the three remaining sites, located below 3400 mbsl (around 1 g cm$^{-2}$ ka$^{-1}$). To visualise long-term trends, we subtracted at each site the mean values of CaCO₃ AR and SR (Fig. 4). All sites show a prominent trend of increasing SR, beginning in the late Miocene (8 Ma ago) (Fig. 4), which is known to reflect increasing amount of clastic material transported from the Amazon Fan (Curry et al., 1995; Pälike et al., 2006b; Bickert et al., 1997; Harris et al., 1997; Shackleton and Crowhurst, 1997). The CaCO₃ AR, on the contrary, does not show any obvious temporal trend (Fig. 4), indicating that the

increase in SR is compensated by decreased carbonate content in the sediment. Instead, the CaCO₃ AR record at all Ceara Rise sites show a pervasive short-term, likely orbital, variability, with substantial magnitude (Curry et al., 1995).

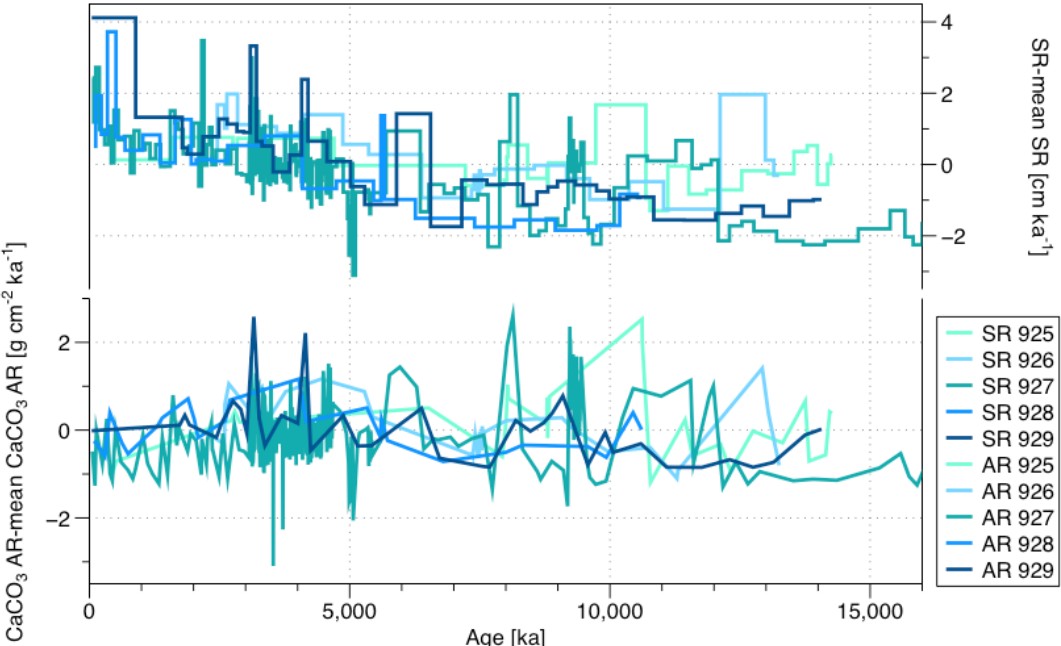


**Figure 4.** a) Sedimentation rate from which the average rate has been subtracted and b) CaCO₃ accumulation rate from which the average CaCO₃ AR has been subtracted, both for the five sites of Leg 154 over the last 16 Ma.

### 3.2 Age models for target intervals at ODP Site 927

### 3.2.1 Pleistocene




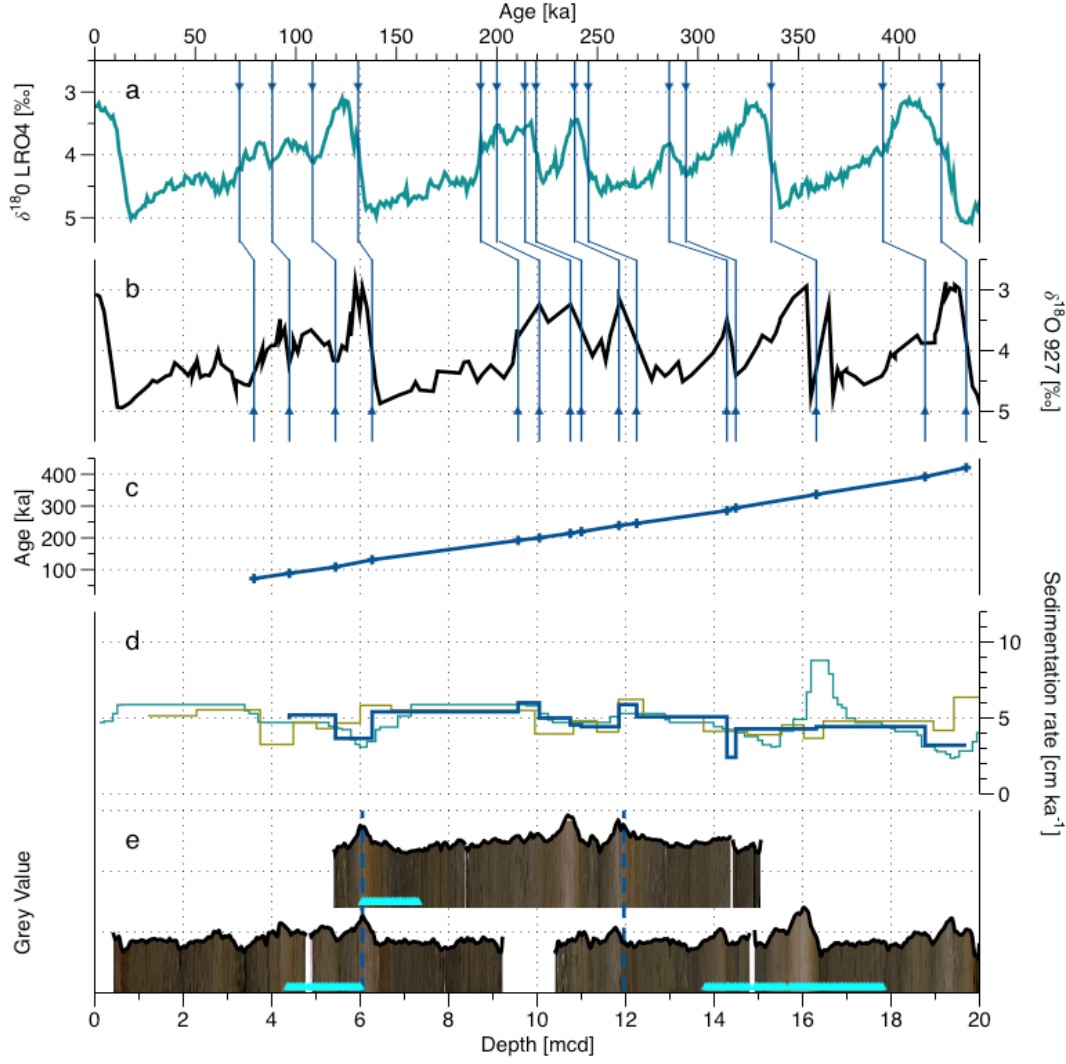

**Figure 5.** Depth - age correlation for the Late Pleistocene, cores 154 927 A1H, 154 927 B2H and 154 927 A2H – following the splice –, with a) the Lisiecki and Raymo, (2005) $\delta^{18}$O stack; b) the local $\delta^{18}$O record (Wilkens et al., 2017, modified from Bickert et al., 2004); c) the age - depth record with control points; d) the SR from Wilkens et al. (2017) (green), the SR using the LRO4 depths and ages for site 927 (blue-green) and the SR defined in this study (blue); e) the core images (Wilkens et al., 2017) and grey value record for the three cores of the spliced used, the position of the switch from one core to the other in the splice (dashed lines) and the position of the samples used in the present study in the cores (blue triangles).

The Pleistocene interval in the studied core has a high-resolution age model based on benthic oxygen isotope data (Bickert et al., 2004) that were incorporated in the benthic stack of Lisiecki and Raymo (2005) who had added a constant 4-5 ka lag to take in account the delay in the $\delta^{18}$O data (ice volume inertia) with respect to the insolation forcing (Lisiecki and Raymo, 2005). However, Wilkens et al. (2017) revised the splice for this site (the way individual core segments are aligned), which means the age model in Lisiecki and Raymo (2005) has to be validated. To this end, we first checked the new alignment of the



individual cores by generating high-resolution sediment colour (grey value) curves from the core images presented by Wilkens et al. (2017) (Fig. 5e). The grey value curve was extracted using the ImageJ software and calculated from RGB images using the NTSC formula (Rasband, 1997) with values averaged across the entire core width perpendicular to the core axis and the resulting noisy curve was smoothed as first component of the singular spectrum analysis (SSA) obtained with Analyseries software (Paillard et al., 1996). This curve was used to compare the overlapping parts of the cores spanning the last 400 ka,

validating the alignment by Wilkens et al. (2017), which we thus adopt without modification. For the age model, we carried out a manual tuning of the 927 $\delta^{18}$O data (Bickert et al., 2004) using the new composite depth by Wilkens et al. (2017) to the LR04 stack (Lisiecki and Raymo, 2005). The tuning was based on the identification of all unambiguously recognisable $\delta^{18}$O maxima and times of fastest sea-level change (Fig. 5a). The resulting SR are indeed more similar to those inferred from the age model by Wilkens et al. (2017) than those implied by the age model for the site as implemented in the LR04 stack (Lisiecki

and Raymo, 2005).

### 3.2.2 Pliocene

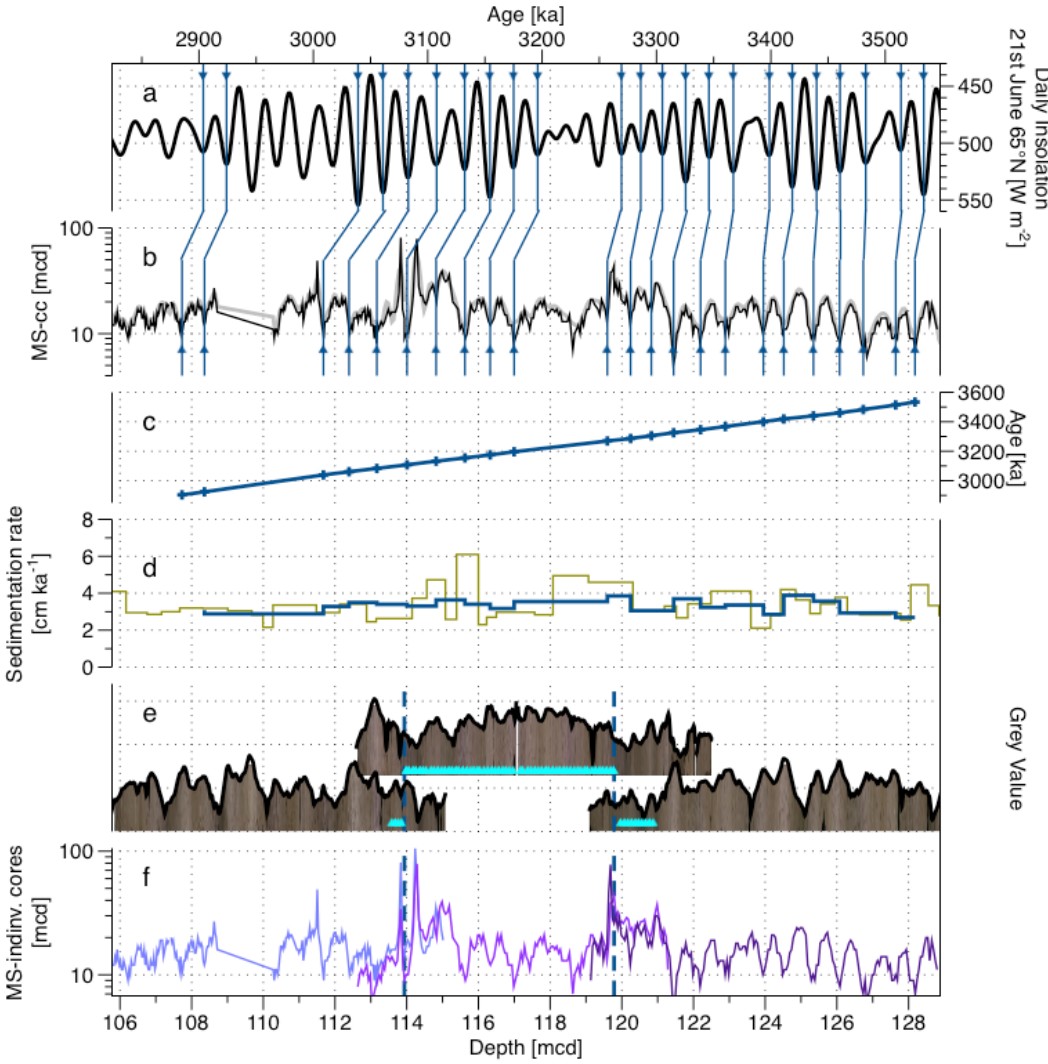



**Figure 6.** Depth - age correlation for the Pliocene interval across cores 154 927C 11H, 154 927A 12H and 154 927B 13H. a)
the daily Insolation 21st of June at 65°N record (Laskar et al., 2004); b) the MS record and MS smoothed record according to
the splice presented in this study; c) the age - depth record with control points; d) the SR from Wilkens et al. (2017) age model
(green) and from this study (blue); e) the core images for the cores of the splice from the ones the samples are from (Wilkens
et al., 2017) and the grey value record extracted from it, plus step from one core to the other in the splice (dashed lines) and
position of the samples used in this study in the cores (blue triangles); f) MS record for the individual cores (Curry et al., 1995;
Wilkens et al., 2017) and steps from one core to the other in the splice (dashed lines).

For the Pliocene interval, the first step has been to validate the core alignment. First, we generated a grey value curve (Sect.
3.2.1) but noted that this signal is weaker and shows many idiosyncratic features among the overlapping parts of the cores
from the individual holes. Therefore, we decided to carry out the tuning on the magnetic susceptibility (MS) signal as done by
Shackleton et al. (1999), which was also measured in all cores (Curry et al., 1995). Magnetic susceptibility shows a distinct
signal in this part of the sediment sequence, which can be used for tuning (like it has been used at Site 926) but for this, it must
be in alignment across the individual core segments. The alignment revealed that the existing splice by Wilkens et al. (2017)
has to be adjusted for the purpose of tuning in this interval (Fig. S2.) by a shift of the core 927 C 11 H by 2 cm shallower, a
shift of the core A 12 H by 15 cm deeper and a shift of the core B 13 H by 9 cm deeper in the splice compared to the spliced
MS record of Wilkens et al. (2017). Otherwise, the construction of the spliced record remained the same, retaining the same
depths where the signal from one core switches to a signal from the adjacent core. These depths are indicated by dashed lines
across the overlapping sections of the cores (Fig. 6d). The spliced MS signal (Fig. 6b) has then been tuned with the daily
insolation on 21st of June 65°N, showing the best pattern of influence from both obliquity and precession (Laskar et al., 2004)
(Fig. 6a). This is possible because the existing age model of Wilkens et al. (2017) is sufficiently precise to provide a specific
tuning target age interval, as confirmed by similar modulation of the insolation target and of the spliced MS record. The tuning
has been done by correlating recognisable 23 MS minima to insolation maxima for this interval, using the Analyseries software
(Paillard et al., 1996), assuming the signals are in antiphase without lag (Wilkens et al., 2017; Zeeden et al., 2013, 2015). As
MS minima are easier to identify than the MS maxima, we here prefered to work with MS minima and Insolation maxima
instead of MS maxima and insolation minima as in Zeeden et al. (2013).

**3.2.3 Miocene**







**Figure 7.** Depth - age correlation for the Mid-Miocene, core 154 927 A33H. a) $\delta^{13}$C loess smooth composite record (from sites U1337 and U1338, Westerhold et al., 2020) and $\delta^{13}$C corrected measured in Miocene samples from this study (*O.umbonatus* in pink, *C. mundulus* in green and *C. wuellerstorfi* in blue, the line corresponds to the average value); b) $\delta^{18}$O loess smoothed composite record (from sites U1337 and U1338, Westerhold et al. (2020) and $\delta^{18}$O corrected measured in Miocene samples from this study (*O.umbonatus* in pink, *C. mundulus* in green and *C. wuellerstorfi* in blue, the line corresponds to the average value); c) Obliquity and E+T-P records (Laskar et al., 2004); d) SSA of the grey value record extracted from the core image - corrected from the light bias- e) age - depth and control points; f) SR from Shackleton et al. (1999) age model (green), from nannofossils events (Curry et al., 1995; Pälike et al., 2010; Wilkens et al., 2017) (dark blue) and from this study (blue); g) $\delta^{18}$O corrected measured on Miocene samples from this study against depth (*O.umbonatus* in pink, *C. mundulus* in green and *C.*





*wuellerstorfi* in blue, the line corresponds to the average value); h) core image (Wilkens et al., 2017), smoothed grey value record and position of the samples for this study in the core (blue triangles).

The existing age model for the Miocene interval by Shackleton et al. (1999) is based on a combination of orbital tuning and biostratigraphy. It presents a distinct shift in the SR around 330 mcd (Fig. 7c), dominating the carbonate AR record for the studied period. There does not seem to be any distinct shift in the physical properties at that depth (Curry et al., 1995), and we therefore felt compelled to test the possibility that the singular change in SR does not correctly represent the changes in the sedimentation at this site. Since the studied interval is within one core segment, we tested whether a "nested" tuned age model

can be developed, allowing a more precise estimation of the variability in the SR. As in this part of the sediment sequence the MS was not the dominant signal, so we have made the choice to work with both the sediment colour and the stable isotopes, to have two independent markers for this age model (analyses ran for the purpose of this study, Sect. 3.2.1. And 2.4.).

To have an independent estimation of the SR, we also evaluated the biostratigraphy from the shipboard (Curry et al., 1995) with revised mcd (Wilkens et al., 2017) and revised biomarker ages GTS 2020 (Raffi et al., 2020). Three biostratigraphic

markers have been evaluated: *LAD Sphenolithus heteromorphus*, *LAD Helicosphaera ampliaperta* and *LAD abundant Discoaster deflandrei*. The combination of these markers gives us two SR options. Using LAD *H. ampliaperta* (the less reliable marker according to Raffi et al., 2020), in combination with LAD *D. deflandrei*, gives a SR of 1.65 cm ka$^{-1}$. Alternatively, considering LAD *S. heteromorphus*, which is recorded in the core further from the studied interval but is considered more reliable according to Raffi et al. (2020), in combination with LAD *D. deflandrei*, gives a SR of 1.11 cm ka$^{-1}$.

A sediment colour proxy was generated for the studied core (Sect. 3.2.1) (Fig. 7d). Due to the light appearance of the sediment composing this core and the way the pictures have been taken onboard (picture on the different 1.5 metres sections with a centred camera and centred white source of light), there is a strong 1.5 metres induced light cyclicity in the original light images (Curry et al., 1995; Wilkens et al., 2017). To reduce this bias, the core images were adjusted for the edge effect using the lighting correction function inside the Code for Ocean Drilling Data (CODD, Wilkens et al., 2017) (Fig. 7d). For the

identification of the cyclicity in the core, we carried out spectral analyses on the corrected grey value curve using the Multi-taper Method (MTM) (carried out using astrochron package on R, Meyers, 2014). This revealed three broad but distinct peaks for the frequencies 0.48 (period: 2.08 m), 0.7 (period: 1.43 m) and 1.4 (period: 0.71 m). Applying the two alternative biostratigraphy-derived SR reveals that the most distinct 71 cm cycles could represent obliquity, when the SR of 1.65 cm ka$^{-1}$ is applied. Finally, we used the $\delta^{18}$O record to define the exact temporal window of the sampled interval and confirm the

assumed cyclicity, by matching the isotopic signal to the Westerhold et al. (2020) stable isotopes loess smooth record $\delta^{18}$O curve as a target. The new isotopic curve reveals a prominent minimum, which corresponds to the 15.6 Ma event, but the older 16.0 Ma isotopic minimum (also seen in carbon isotopic record of the target) is not recorded, indicating that the sampled interval spans less than 400 ka and the average SR must be >1.2 cm ka$^{-1}$. Beyond the two prominent minima, the isotopic curve does not allow direct tuning because of low resolution and ambiguous shape of the target (Fig. 7). Therefore, we used the

alignment to the younger isotopic minimum and the E+T-P signal as a target curve (taking in account the eccentricity, the obliquity and the precession) (Fig. 7a) (Laskar et al., 2004), to tune prominent light minima with E+T-P minima (and Obliquity minima) (Shackleton et al., 1999; Zeeden et al., 2013) (Fig. 7a and b). This tuning has been then verified by plotting the stable isotope (both $\delta^{18}$O and $\delta^{13}$C) record using the new given ages, and comparing it to the existing stable isotopes loess smooth records from Westerhold et al. (2020) (Fig. S3.).

**3.3 High resolution records of carbonate content and carbonate accumulation rates at ODP Site 927**





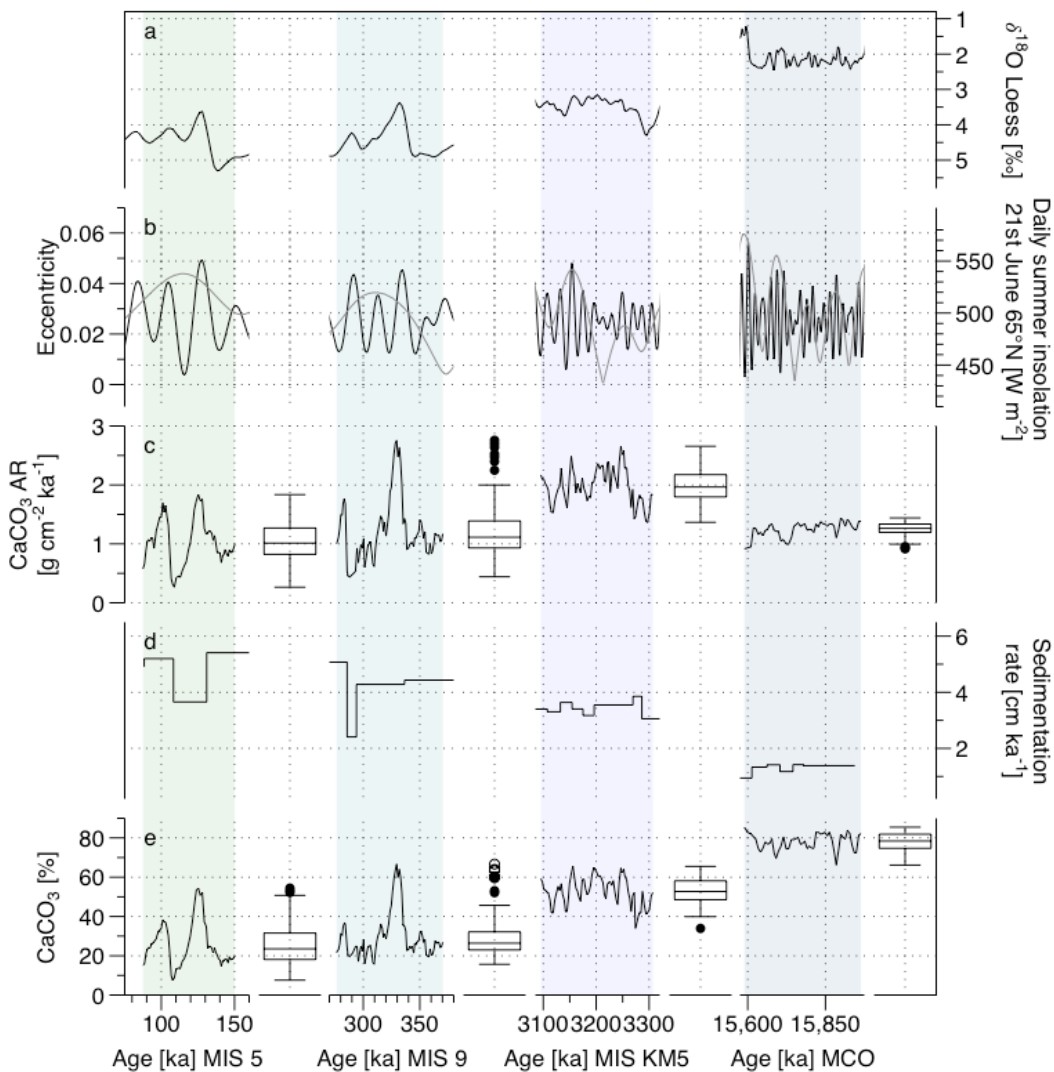

**Figure 8.** a) Stable isotopes $\delta^{18}$O local and global (Bickert et al., 2004; Westerhold et al., 2020); b) orbital parameters: eccentricity and daily summer insolation 21$^{st}$ of June at 65°N (Laskar et al., 2004); c) CaCO$_3$ AR and boxplot for each interval;

d) SR and e) CaCO$_3$ % in the dried bulk sediment and boxplot for each interval.

The new carbonate content analyses are based on 261 measurements, yielding values comparable to existing low-resolution measurements, confirming decreasing carbonate content throughout the Neogene (due to dilution by clastic sediments from Amazon fan, (Curry et al., 1995; Bickert et al., 1997; Harris et al., 1997) and indicating particularly strong variations in the

Quaternary (Fig. 8). In combination with the new high-resolution SR data (Fig. 8), these measurements provide records of sub-orbital variability in CaCO$_3$ AR across the four intervals, showing orbital-scale variability exceeding the differences in mean CaCO$_3$ AR among the intervals (Fig. 8).

The comparison between the highly resolved record for the four periods of interest (Fig. 8c, 8d and 8e) and the environmental parameters (Fig. 8a and 8b) is highlighting the good correlation - in terms of phase and amplitude - between the CaCO$_3$ AR

(reflecting the pelagic carbonate production) and the insolation at 65°N signal for the two warm interglacials observed. For the





MIS 5 and the MIS 9 warm interglacials, there is a strong correlation between the $CaCO_3$ % and $CaCO_3$ AR, ($R^2$ of 0.86 and 0.93) (Fig. 8 and Fig. 9). At the same time, the SR is reaching high values (3 to 5 cm ka$^{-1}$), independently of $CaCO_3$ AR changes, indicating the role of another component than the pelagic carbonate production influencing the SR. In contrast, during the MIS KM5 and the MCO, the $CaCO_3$ AR is driven by both the carbonate content and the SR (Fig. 9).


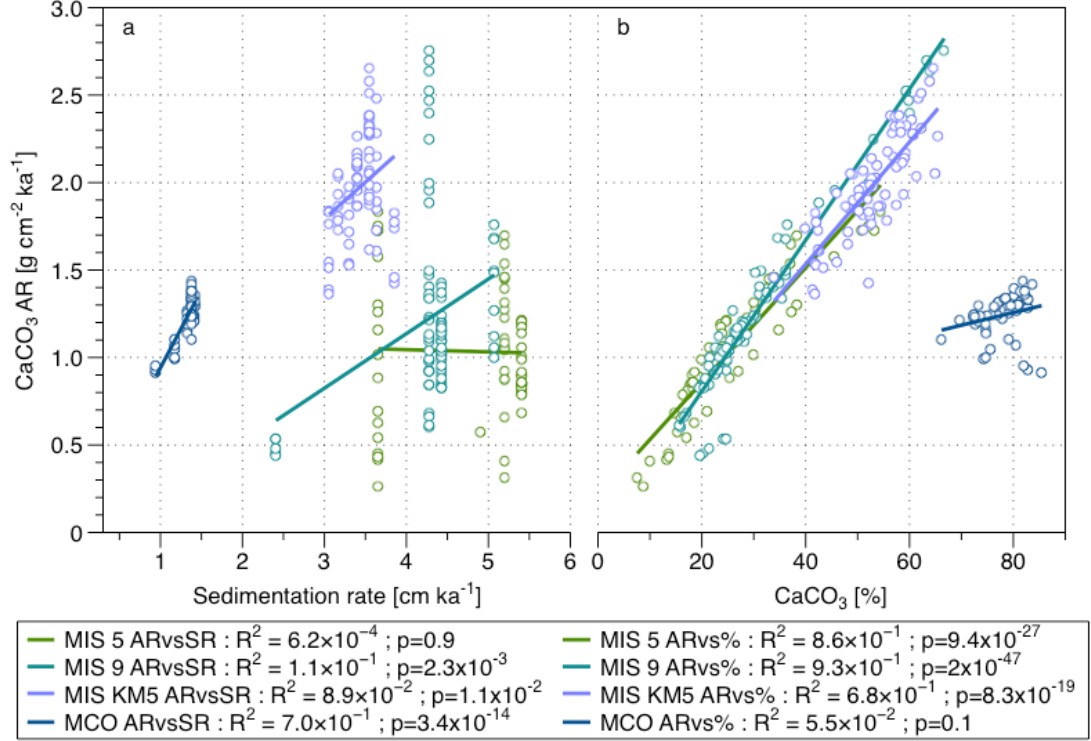

**Figure 9.** a) Relationship between the $CaCO_3$ AR and the SR and b) relationship between the $CaCO_3$ AR and the $CaCO_3$ % for the four periods of interest.


Furthermore, the slope of the relationship between carbonate content and carbonate AR appears to decrease with time, indicating that the earlier in the record, the less the carbonate percent is influencing the carbonate AR.





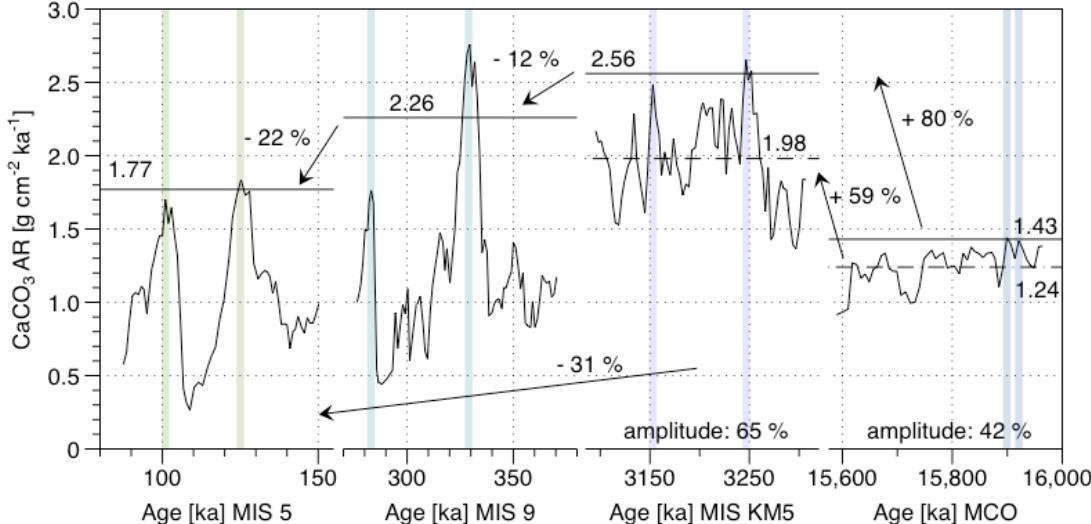

**Figure 10.** CaCO₃ AR for each period of interest, average value between two maxima per period (for Quaternary and Pliocene), average value taking in account maxima and minima for the Pliocene and the Miocene, and quantification of change within (amplitude)/between each of the periods of interest. The shaded areas are underlying the maxima values of CaCO₃ AR used for the quantification of change between the time intervals calculation.

When we look at the trend of the highest values reached on long geological time scale from mid-Miocene to Eemien (Fig. 10), we observe a 31 % decrease of CaCO₃ AR from the Pliocene (highest value) to the Pleistocene MIS 5 (lowest value), excluding dissolution intervals in the Pleistocene. Taking in account the average value of the MCO and MIS KM5, we found an increase of the pelagic carbonate production of 59 % from Miocene to Pliocene MIS KM5. If we now take in account the maxima values for the Quaternary and Pliocene MIS KM5, we observe a decrease of 12 % from the Pliocene MIS KM5 to Pleistocene and a decrease of 22 % from Pleistocene MIS 9 to Pleistocene MIS 5.

Looking at the amplitude of the variability within the Pliocene and Miocene interval, we found higher value in the Pliocene (65 %) compared to the average of the period (1.98 g cm⁻² ka⁻¹) than in the Miocene (42 %) compared to the average (1.24 g cm⁻² ka⁻¹).

## 4 Discussion

### 4.1 Carbonate preservation

During the Pleistocene, the carbonate AR at Site 927 is driven only by the carbonate content, indicating that the signal is affected by dissolution. This is confirmed by the presence of very low values of carbonate content and carbonate AR during the cold intervals in the Pleistocene, in phase with the $\delta^{18}$O and insolation signal, indicating a relationship to changes in deep water circulation, confirming the conclusions by Bickert et al. (1997). Because the Ceara Rise sites became periodically affected by the more corrosive Antarctic bottom water only after the initiation of the North Hemisphere glaciation (Liebrand et al., 2016; Harris et al., 1997; Pälike et al., 2006a), the studied Pliocene and Miocene intervals are not affected by dissolution. Throughout the entire studied interval since the Miocene (Fig. 3), the shallowest cores (925 and 927) record higher CaCO₃ AR values than the deeper ones. This indicates that these sites likely remained above the lysocline (at present located around 4000 m) (Curry et al., 1995; Bickert et al., 1997; Frenz et al., 2006; Gröger et al., 2003a, b), and that the CaCO₃ AR signals recorded





at these sites record changes in pelagic carbonate production. It provides an ideal target for orbital tuning, with existing age models indicating continuous sedimentation (Wilkens et al., 2017; Shackleton et al., 1999). Indeed, as expected from the overall stratigraphy and paleoceanography of the Ceara Rise sites (Curry et al., 1995; Frenz et al., 2006; King et al., 1997), the new carbonate content (Fig. 8e) and CaCO$_3$ AR (Fig. 8c) records from Site 927 show strong minima during cold intervals

(indicated by δ$^{18}$O record) which is consistent with the shoaling of the corrosive AABW (Miller et al., 2012; Harris et al., 1997), causing dissolution at shallower depth (Gröger et al., 2003a, b). In contrast, there is no dissolution observed during the MIS KM5 and the MCO in the new records from Site 927 and also the maxima in carbonate AR and carbonate content in the sediment during the Quaternary do not appear to be affected by dissolution. Therefore, whilst we cannot use the Quaternary variability in the carbonate AR to estimate the orbital-scale variability in pelagic carbonate production, we can use the

interglacial maxima (Fig. 10) to estimate pelagic carbonate production during the Quaternary.

**4.2 Orbital variability in Pliocene and Miocene**

Since the dissolution is not the main driver of the observed changes, and the pelagic carbonate is the main component of the carbonate fraction of the sediment (Curry et al., 1995), we here observe the changes in the export flux of pelagic biogenic carbonate. Following this argument, the new record from Site 927 reveals that pelagic carbonate production (assessed by the

pelagic carbonate AR) in the equatorial ocean (avoiding large amplitude temperature changes) has changed on geological time scale by a factor of two and on orbital time scales by up to 50 %. The presence of orbital-scale variability in pelagic carbonate production is an interesting phenomenon, which requires further analysis. First, we tested whether or not this variability is periodic, i.e. whether the underlying changes in pelagic carbonate production responded to orbital forcing. Such analysis is possible, because the studied intervals have been tuned to the orbital target using parameters other than carbonate content (Fig.

6 and 7). Multi-Taper method (MTM) spectra derived with the Astrochron package on R (Meyers, 2014) (Fig. 11) highlight significant periodicity close to the precession band for the MIS KM5 and periodicities in the obliquity and 100 ka eccentricity bands for the MCO. This implies that during both intervals, the pelagic production likely varied in response to orbitally-driven environmental factors, such as insolation (light intensity for phytoplankton, Cavaleiro et al., 2018) or nutrient availability due to changes in upwelling (Cavaleiro et al., 2020). Interestingly, the dominant periodicities appear different between the Pliocene

and Miocene. Next, we asked whether or not the observed periodicities in carbonate AR are coherent with the actual insolation, obliquity and eccentricity signals. This is possible because the underlying age models have been tuned such that they should preserve the correct phase relationship with the orbital forcing (Fig. 6 and 7). To this end, we carried out cross Blackman-Tukey (BT) analyses using the Analyseries software 2.0 (Paillard et al., 1996). The results (Fig. 12) indicate a coherence with insolation in the precession band as well as with the 41 ka obliquity for the MIS KM5. In both cases, the coherence occurs in

phase. In contrast, for the MCO, we observe a coherence at 41 ka with the obliquity periodicity and at 100 ka with eccentricity, but in both cases, the coherence is anti-phased. That the pelagic carbonate production is responding to an eccentricity paced periodicity (Fig. 11 and 12) is interesting, as eccentricity was not the main driver of the Earth climate signal (Westerhold et al., 2020; De Vleeschouwer et al., 2020). On the other hand, the carbon cycle in the Miocene appears to show eccentricity pacing (Holbourn et al., 2007, 2018; De Vleeschouwer et al., 2020; Raitzsch et al., 2020), and our results indicate that pelagic

carbonate productivity may play a role in modulation of this cyclicity. Also, we note that the discovery of eccentricity forcing pelagic carbonate production in the Miocene and a shift towards obliquity and precession forcing in the Pliocene is consistent with the observations from mid-latitudes by Drury et al. (2020) and the modelling study by Vervoort et al. (2021) provides potential mechanisms on how the eccentricity and obliquity frequencies in carbonate production may arise despite the dominance of the precession frequencies in the forcing.






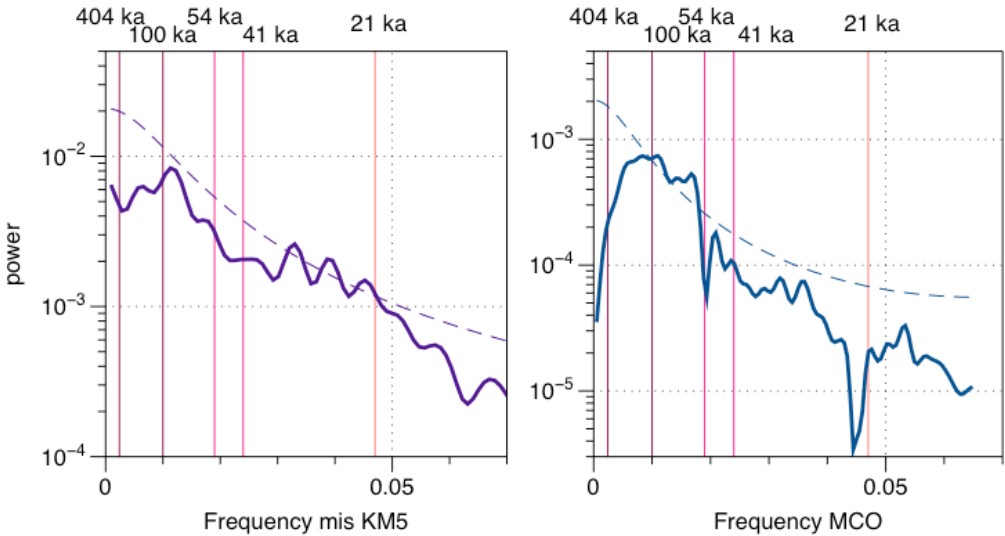

**Figure 11.** MTM spectral analysis of the CaCO$_3$ AR record through time (Meyers, 2014). The dashed lines represent the 95 % significance level. The pink shadows correspond to the orbital periodicities (eccentricity 404 ka and 100 ka, obliquity 54 ka and 41 ka and precession 21 ka).

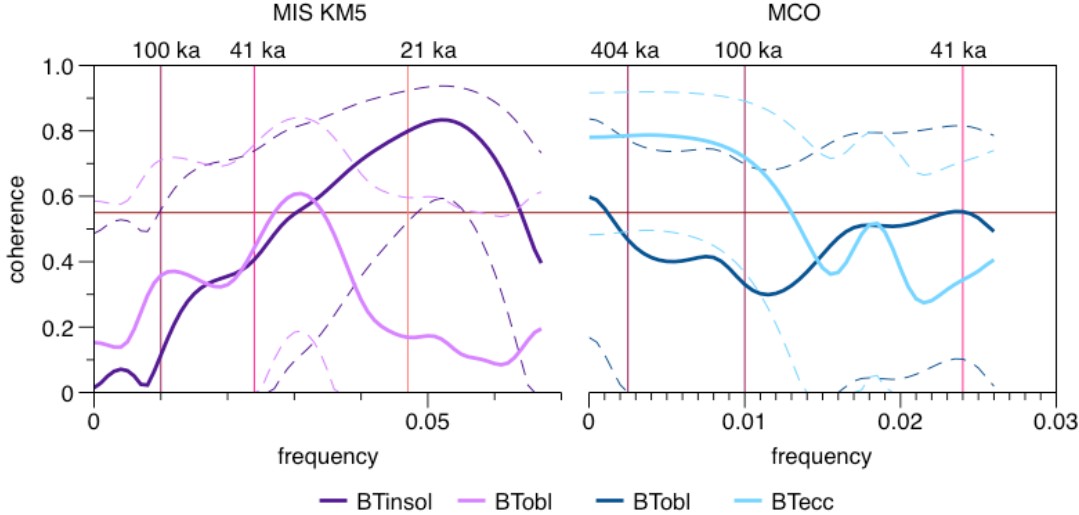

**Figure 12.** Coherence diagram BT cross correlation (Paillard et al., 1996) between the CaCO$_3$ AR and the orbital parameters (Laskar et al., 2004). The horizontal red line corresponds to the non zero coherence for a significance level of 90 %. The pink shadows correspond to the orbital periodicities (eccentricity 404 ka and 100 ka, obliquity 54 ka and 41 ka and precession 21 ka).





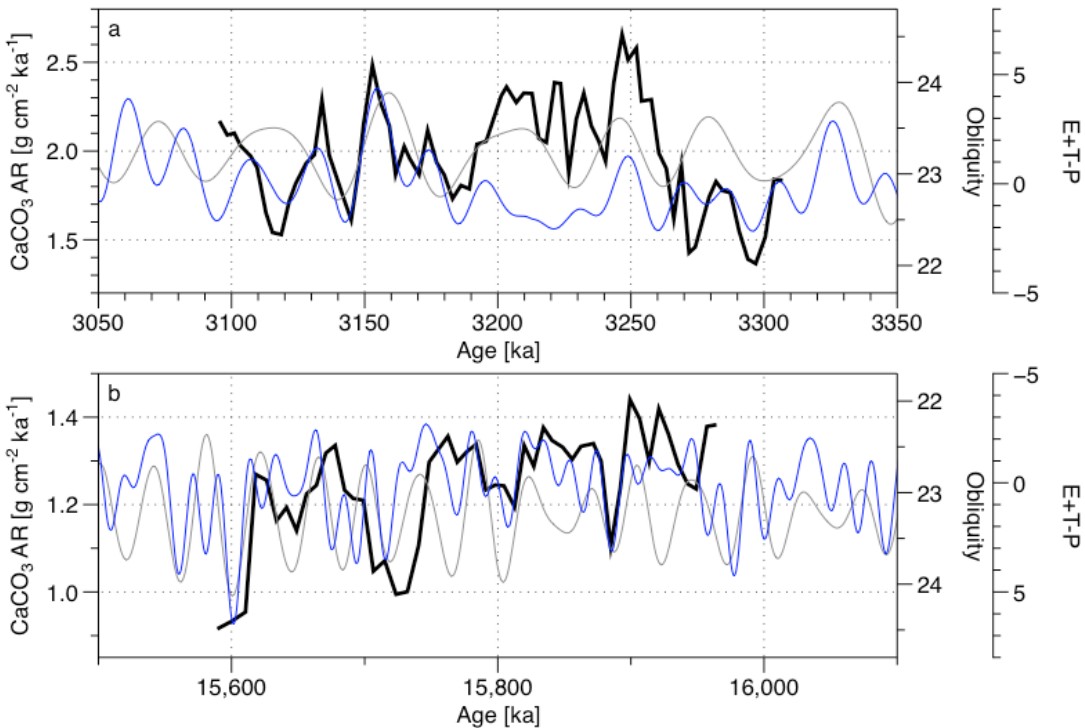

**Figure 13.** Comparison of the CaCO$_3$ AR (black) with both the obliquity (grey) and the E+T-P orbital records (blue) for a) the MIS KM5 and b) the MCO.

Finally, we consider the apparent shift in the phase in the relationship between orbital forcing and the CaCO$_3$ AR record between the Pliocene and Miocene. This relationship implied by the cross-spectral analysis is clearly visible in the raw data (Fig. 13) and we consider it unlikely that it is due to tuning artefacts. We note that the Miocene record ends with a strong and distinct minimum in the oxygen isotope record, which provides a strong constraint on the phase relationship between the youngest carbonate AR and obliquity cycle. These show an opposite phase relationship to that observed during the Pliocene. This could be explained by a change of the carbonate production response to the insolation changes between the Pliocene and the Miocene. Indeed, the production of different pelagic calcifiers could be promoted by a decreased mean annual insolation at equatorial latitude (with high E+T-P and high obliquity) during the Pliocene compared to the Miocene, when the pelagic carbonate calcifiers appears to be promoted by a higher mean annual insolation at equatorial latitude (with low E+T-P and low obliquity). We can then expect a higher weight of the foraminifera (non photosynthetic) in the carbonate production balance during the Pliocene and a higher weight of the coccolithophores (doing photosynthesis) in the carbonate production balance during the Miocene. This is coherent with the climate-carbon cycle changes occurring between the Miocene and the Pliocene, highlighted by De Vleeschouwer et al. (2020), who found changes in the phase relationship of δ$^{18}$O and δ$^{13}$C before and after 6 Ma.

### 4.3 Long-term trend (differences between periods)

Because of the observed changes in what appears to be carbonate production among the studied intervals and especially within the studied intervals, we conclude that tropical pelagic calcifiers responded to environmental or biotic forcing on orbital cycles,










as well as to long-term shifts in climate and/or ocean chemistry. In other words, either the production, the community composition or the biomineralisation of the tropical pelagic calcifiers responded to local changes in light, temperature and nutrients delivered by upwelling, which followed orbital cycles, as well as to long-term shifts in climate and/or ocean chemistry.

The inferred changes in pelagic carbonate production on both time scales are sufficiently large that when extrapolated on a global scale, they could have played a role in the regulation of the carbon cycle. For example,Boudreau et al. (2018) estimated that changes in global pelagic carbonate production on the order of 10 % would be sufficient to affect the marine carbon cycle on time scales from year to million years. Whereas the drivers of the orbital-scale variability could be plausibly attributed to changes in local oceanic parameters affecting primary production, the causes of the long-term shifts require another

explanation.

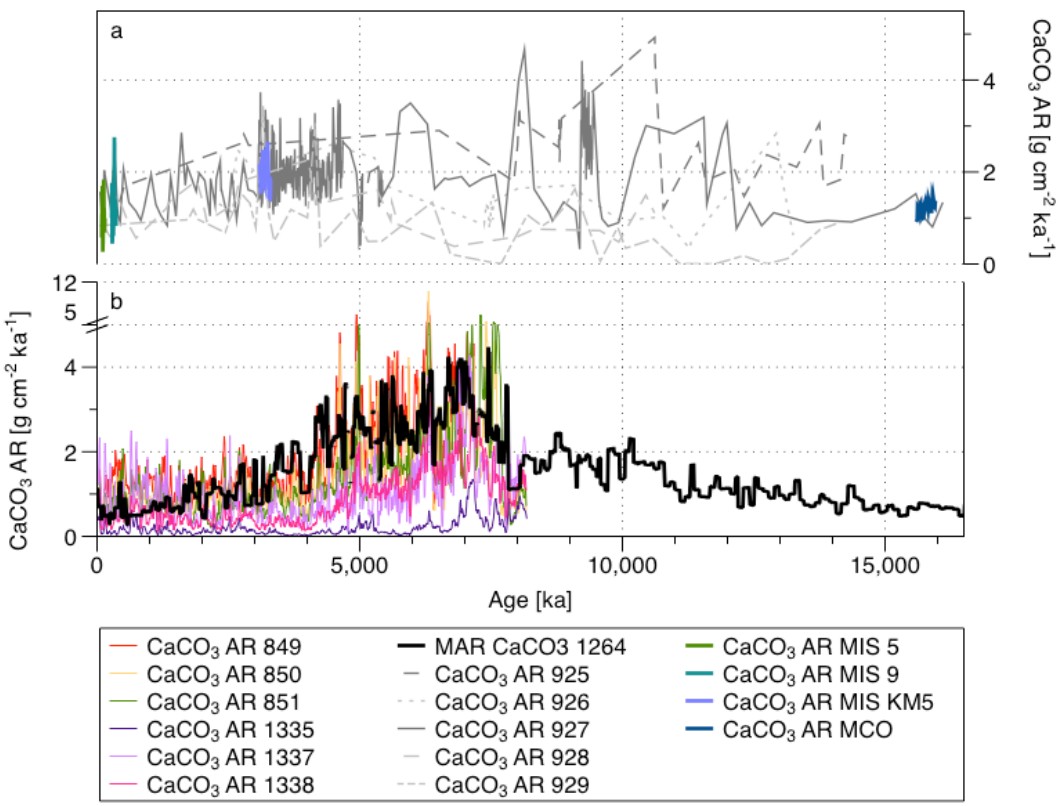

**Figure 14.** Comparison of the CaCO₃ AR at a) the 5 Ceara Rise sites (Sect. 3.1., in Grey) and CaCO₃ AR at high resolution
(this study; in colours) and b) CaCO₃ AR record in the equatorial Pacific (colours) (Lyle et al., 2019) and South Atlantic Ocean (black) (Site 1264, Drury et al., 2020).

First, we note that the observed carbonate AR at the Ceara Rise is coherent with the observations made by Lyle et al. (2019) in the Equatorial Pacific Ocean and Drury et al. (2020) in the South Atlantic Ocean (Fig. 14), as well as with the recent results
by Sutherland et al. (2022) from the South Pacific. Our record is showing similar absolute values (an AR between 0 and 5 g cm⁻² ka⁻¹) and a largely similar overall trend with highest values in the late Miocene/early Pliocene and similar values in the early Miocene and Quaternary. Clearly, the overall trend of carbonate accumulation at the Ceara Rise supports the existence



of a late Miocene carbonate maximum also under tropical conditions. Interestingly, our observations from the Ceara Rise also support the conclusion from Sutherland et al. (2022) that there does not appear to be any strong relationship between pelagic
carbonate production and global $CO_2$, other than the fact that the lowest carbonate AR in both their and our records are observed during the MCO with presumably highest $CO_2$.

## 5 Conclusion

A compilation of carbonate accumulation rates for the five sites of the Leg 154 in the Western Equatorial Atlantic Ocean documents a distinct increase in sedimentation rates from Miocene to Quaternary, but the carbonate accumulation rate
remained relatively stable. The two shallowest sites consistently record higher carbonate accumulation rates, confirming observations of good carbonate preservation during Quaternary interglacials and throughout the Pliocene and Miocene. This means that the observed changes in carbonate accumulation at these sites should reflect changes in pelagic carbonate production.

To analyse long-term and orbital-scale patterns of pelagic carbonate production variability, we generated new data for four
periods at Site 927. This site was selected because the sediments preserve periodic signals in sediment properties and/or benthic oxygen isotope values that can be tuned to orbital cyclicity, allowing us to construct an age model that can resolve changes in carbonate accumulation rate on orbital time scales.

We found that carbonate accumulation rate, as a proxy for pelagic carbonate production in the tropical Atlantic, exhibited both long-term changes and a pervasive orbital-scale variability. We observe a 31 % decrease of carbonate accumulation rate from
the Pliocene MIS KM5 to the Pleistocene interglacial MIS 5, but 59 % higher values for the Pliocene warm period than for the Miocene climatic optimum. On the orbital time scale, the Quaternary signals are overprinted by precession-insolation forcing on deep water circulation, causing dissolution. However, concerning the Pliocene warm period and the Miocene Climatic Optimum, we observe a persistent variability in carbonate accumulation with an amplitude exceeding that of the long-term mean shifts. We show that the carbonate accumulation rate at low latitude varied in phase with insolation (precession) cycles
during the Pliocene, whereas the Miocene signal is dominated by 100 ka eccentricity cycles, which are exactly antiphased with the carbonate signal.

We conclude that the low-latitude pelagic carbonate production responded strongly to orbital-driven local tropical processes, rather than to secular changes in the global climate or ocean chemistry (like global $CO_2$). The Ceara Rise records are consistent with the existence of a Late-Miocene to Pliocene global carbonate production optimum, but the magnitude of the long-term
change appears smaller than outside the tropics. Instead, orbital-scale variability dominates the record and the inferred magnitude of production changes are potentially sufficient to affect the global carbon cycle through the process of biological compensation (Boudreau et al., 2018).

Our results imply that in the context of the ongoing and projected global change, pelagic carbonate production may be an important variable in the parametrisation of the global marine carbon cycle, especially with regard to the long-term (millennial-
scale) fate of anthropogenic carbon injection. To parametrise the pelagic carbonate production, it remains to be shown whether it changes due to changes in production (population sizes), biomineralisation (amount of carbonate produced per individual) or community composition (shift to more or less calcified taxa).

### Data Availability

Datatables will be made available upon request to the the main author until their online publication on PANGAEA
(https://pangaea.de, last visit: 23[rd] March 2022)

### Author contribution





The conceptualisation has been carried out by all the co-authors. PC generated the data and ran the analyses. All authors contributed to writing the manuscript.

**Competing interests**

The authors declare that they have no conflict of interest.

**Acknowledgments**

This research used samples and data provided by the Ocean Drilling Program (ODP), sponsored by the US National Science
15 Foundation (NSF) and participating countries. This research was supported by the DFG through Germany's Excellence Strategy, Cluster of Excellence "The Ocean Floor—Earth's Uncharted Interface" (EXC-2077, Project 390741603). We thank
Brit Kockisch for assistance with carbonate content analyses and Anna-Joy Drury for providing South Atlantic carbonate data and discussing the results.

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
