# Peer review of "Nature and origin of variations in pelagic carbonate production in the tropical ocean since the Mid Miocene (ODP Site 927)"

_Biogeosciences, 2022_

## Referee Comment (RC2)

This manuscript by Cornuault and co-authors uses carbonate accumulation rates to (1) evaluate long-term changes in pelagic carbonate production since the Middle Miocene, and in particular during four major past warm intervals: the MIS 5, the MIS 9; the PWP and the MCO; (2) characterize the orbital-scale variability and (3) determine whether orbital periodicity forcing carbonate production changed from the Middle Miocene to present. To do that, the authors first compiled existing CaCO3 % from all the Leg 154 sites in the Ceara Rise (equatorial Atlantic Ocean) since the Miocene. Next, they revised the age models of these sites to calculate sedimentation rates and carbonate accumulation rates. Moreover, they generated new $\delta^{18}O$, $\delta^{13}C$ and carbonate content data from Site 927 (Leg 154) for the four selected time intervals and developed updated age models for each time interval to calculate carbonate accumulation rates.

The results show a general increase in sedimentation rates since the late Miocene, but the carbonate accumulation rates did not show a similar clear trend. Additionally, the authors observed that the highest carbonate accumulation rates occurred during the Pliocene. Furthermore, they demonstrate that variations in carbonate accumulation prior to the Quaternary cycles follow obliquity and eccentricity and suggest that this reflect changes in the export flux of pelagic biogenic carbonate. The authors thus propose that the overall carbonate production responded to local changes in light, temperature and nutrients delivered by upwelling, which followed long orbital cycles and long-term shifts in climate and/or ocean chemistry. Finally, they suggest that the observed changes were sufficiently large that could have played a role in the regulation of the carbon cycle and global climate evolution during the Miocene warm climates into the Quaternary icehouse.

The manuscript adds to a growing body of knowledge surrounding the controlling factors regulating the global carbon cycle and global climate evolution. The conclusions are therefore significant. The selection of the four warm time intervals is appropriate, as they represent key warm periods of the late Cenozoic, selected by the international scientific community. The methods are appropriate for the work.

I generally agree with the interpretations, however, as part of my review I have some points the authors should consider/address during revision that I don't think will result in significant changes to the conclusions. In particular, there needs to more evidence to indicate that changes in carbonate AR are not affected by dissolution, especially for Site 927. There also needs to be more discussion on the link between the carbonate production and the driving mechanisms (e.g., light, temperature, nutrients-upwelling processes).

I also have some other minor recommendations and corrections listed below.

With some moderate to major revision this manuscript will represent an important contribution for publication in Biogeosciences.

**Main Review Points:**

1.  a) I found the first part of the discussion section "carbonate preservation" rather weak. Although this does not mean that I necessarily disagree with author's arguments, but as this section is very important for the next parts of the paper, I recommend to provide more evidence indicating that the observed changes are (not) driven by dissolution. A series of Scanning Electron Microscope (SEM) images for instance could be helpful, or/and comparison with other available data (e.g., biogenic siliceous productivity) (maybe add a figure in supplementary material).

    b) Moreover, the authors infer in several places in the text that Leg 154 sites remain either above or below lysocline based on their modern depths. Given that depth is a crucial parameter for dissolution/preservation, I recommend to provide information on the paleodepths of the sites, especially for Site 927 for all studied time intervals.

2. a) The authors propose that changes in light, temperature and nutrients driven by upwelling, forced the observed changes in the export of flux of pelagic biogenic carbonate. These could be plausible mechanisms, but I would like to see a more detailed discussion on this. The authors could use available published data (e.g., SST; Herbert et al., 2016) to back up their hypothesis. Additionally, in Lines 22-23, they state "These results imply that the pelagic carbonate production in the tropical ocean, buffered from large temperature changes, varied…." Are there available data that shows that?

b) I also recommend to include a final figure (conceptual model) summarizing the main conclusions: changes in carbonate AR for the different time intervals, orbital variability, as well as potential mechanisms (e.g., light, temperature and nutrients).

3. In line 239, you state that "the $CaCO_3$ AR, on the contrary, does not show any obvious temporal trend (Fig. 4), *indicating that the increase in SR is compensated by decreased carbonate content in the sediment*". Maybe I'm confused, but when I'm looking for example Site 927 in Figs 3 and 4, I see that increased SR coincide with increased carbonate content between 16 and 3 Ma. Could you explain this better?

4. I'm missing a section in the result part for the new $\delta^{18}O$, $\delta^{13}C$ data generated from this study. Moreover, these data can provide additional information that can help the part of discussion.

5. I recommend to add a section of modern hydrography of the region.

**Minor comments:**

Lines 20-21: "…, but that each interval was characterized by large orbital-scale variability" Although I understand what you mean, reword if possible.

Lines 23-24: "…on orbital time scales similarly or even more than on longer time scales". Rephrase.

Line 71: "plankton had no opportunity to responds to the climate cycles by migration" Add a reference.

Line 80: "… to assess the spatial coherence of long term

Line 111: "…is also characterized as wetter" wetter compared to today? clarify

Line 130: "This aseismic ridge rises several km above…" Give depth

Line 273: "… and times of *fastest* sea-level change..." What do you mean by fastest sea level change?

Line 506: "a *largely similar* overall trend…" I cannot see that - reword this part.

Lines 507-508 you state "*Clearly*, the overall of carbonate accumulation at the Ceara Rise supports the existence of a late Miocene carbonate maximum also under tropical conditions". However, in lines 15-16 you note that there is "a systematic increase in sedimentation rates since the late Miocene, but carbonate accumulation rate does not show a clear trend", which is what your data show. Therefore, these lines in the discussion need rewording.

Lines 515: "… The two shallow sites consistently…." Add sites in a parenthesis to remind them to the reader.

**Figures**

Figure 1: Add scale for bathymetry

Figure 3: This is a nice figure. 3b: I recommend to add also a small key-scale showing the values of the colors.

---

## Author Response (AR1)

**Author's response to the reviews**

We thank the reviewers for the comments they gave on our work, that helped us to review and improve our manuscript.

We have considered all the different reviewer's comments (Reviewer#1 in pink, Reviewer#2 in blue) and modified the text according to the suggestions we have made in the responses to the reviewers. Here, we took back point by point the reviewer's comments and our responses to it, listing the relevant changes made in the manuscript. They are all addressed/discussed as follows (the arrows are our modification to the reviewer comment written above).

All the changes that have been done are visible (highlighted in green) and presented in the "Author's track-changes file" also given with the resubmission of our manuscript.

There needs to more evidence to indicate that changes in carbonate AR are not affected by dissolution, especially for Site 927. There also needs to be more discussion on the link between the carbonate production and the driving mechanisms (e.g., light, temperature, nutrients-upwelling processes).

1. a) I found the first part of the discussion section "carbonate preservation" rather weak. Although this does not mean that I necessarily disagree with author's arguments, but as this section is very important for the next parts of the paper, I recommend to provide more evidence indicating that the observed changes are (not) driven by dissolution. A series of Scanning Electron Microscope (SEM) images for instance could be helpful, or/and comparison with other available data (e.g., biogenic siliceous productivity) (maybe add a figure in supplementary material).

→ The dissolution section has been expanded and restructured, separating the Quaternary part from the Pliocene and Miocene part. Concerning the proof of a good preservation for the Pliocene and Miocene intervals, we added three figures in the supplements in order to support our arguments (we have in the meantime generated for the Pliocene and Miocene data on fragmentation of planktonic foraminifera shells, a commonly accepted proxy for the extent of carbonate dissolution). The first one is the fragmentation index of the <63 µm size fraction along time, for both the Pliocene and the Miocene intervals. The second figure is a plot of the fragmentation index in the <63 µm size fraction versus CaCO$_3$ AR. And finally, the third figure is presenting pictures to document the preservation state of key samples, representing the high and low carbonate accumulation rates for each period, by high-resolution optical images from a digital microscope. As shown, the fragmentation varies, but remains low, indicating no evidence for dissolution and, most importantly, the fragmentation does not correlate with carbonate accumulation rate at all, indicating that the observed changes in carbonate accumulation must reflect processes other than dissolution.

1. b) Moreover, the authors infer in several places in the text that Leg 154 sites remain either above or below lysocline based on their modern depths. Given that depth is a crucial parameter for dissolution/preservation, I recommend to provide information on the paleodepths of the sites, especially for Site 927 for all studied time intervals.

→ We used more explicit data to constrain and quantify a possible effect of this variable. With regard to the paleodepth, we added a paragraph to mention that the variations since mid-Miocene have been negligible. Paul et al. (2000) note that the exact subsidence history is unknown, but assume minimal

subsidence since early Miocene. Similarly, sea-level differences among Quaternary interglacials and Pliocene and Miocene were likely in the order of 10s of metres. Therefore, the largest changes in paleodepth would have been due to sediment cover, which would make the studied mid-Miocene interval about 300 m deeper compared to the present one.

2. a) The authors propose that changes in light, temperature and nutrients driven by upwelling, forced the observed changes in the export of flux of pelagic biogenic carbonate. These could be plausible mechanisms, but I would like to see a more detailed discussion on this. The authors could use available published data (e.g., SST; Herbert et al., 2016) to back up their hypothesis. Additionally, in Lines 22-23, they state "These results imply that the pelagic carbonate production in the tropical ocean, buffered from large temperature changes, varied…." Are there available data that shows that?

→ The sentence has been modified to make it clear that we list these parameters as options, but cannot at present resolve which was more important for the observed changes in carbonate production. In terms of the claim that the tropical ocean was buffered from large temperature changes, we provided references showing modest SST variation compared to higher latitudes and highlight the fact that we mean buffered compared to higher latitudes. Low-magnitude tropical SST variability in the Atlantic in the Pliocene and the Miocene was reported by Herbert et al. (2016) and Curry et al. (1995).

2. b) I also recommend to include a final figure (conceptual model) summarizing the main conclusions: changes in carbonate AR for the different time intervals, orbital variability, as well as potential mechanisms (e.g., light, temperature and nutrients).

5. I recommend to add a section of modern hydrography of the region.

→ We provided a more extensive description in the introduction at the place where we introduce the site (line 129).

Line 85: add age range for each of the four intervals examined.

Figures: I suggest to give ages in Ma rather than in ka, so one can get rid of all the zeros.

→ The ages of the studied intervals have been added in brackets in the text;
→ Since we have two intervals in the Quaternary, we stayed in ka;
→ We investigated four intervals, which occurred during the four listed periods (not the four listed periods entirely). Therefore, we did not only add the exact time in brackets for the four intervals, but also specified the above in the text.

Line 234: How about productivity changes? Sites 928 & 929 are farer to the coastline compared to other sites. Can the higher distance to the Amazon fan result in lower nutrient deliver and thus lower biological production at those sites?

→ A brief statement explaining the possible role of the distance from the Amazon Fan and productivity has been added in the revised manuscript.

R#1 : Line 239: "The CaCO3 AR, on the contrary, does not show any obvious temporal trend" I do not agree. In my opinion the CaCO3 AR generally increases until ~4 Ma, and then it slightly decreases.

R#2 : 3. In line 239, you state that "the CaCO3 AR, on the contrary, does not show any obvious temporal trend (Fig. 4), indicating that the increase in SR is compensated by decreased carbonate content in the sediment". Maybe I'm confused, but when I'm looking for example Site 927 in Figs 3 and 4, I see that increased SR coincide with increased carbonate content between 16 and 3 Ma. Could you explain this better?

→ We did not intend to insinuate that there is no trend in the CaCO$_3$ AR at all, but wanted to highlight that the observed changes are much less obvious than the strong increase in overall sedimentation rate. The statement has been improved accordingly.

Lines 272-273: what do the authors mean with "fastest sea-level changes"? Do they mean that they interpret the Site 927 d18O record as reflecting sea-level changes? If so this needs to be stated and the motivation for such an interpretation needs to be explained.

→ Indeed, the benthic stable oxygen isotope record from Site 927 published by Bickert et al. (2004) during the Quaternary reflects chiefly sea-level changes. This is why it could be included in the L&R stack and why it should be interpreted and used for correlations as such. A statement has been included at this place.

Line 342: I think it is necessary to add a figure, perhaps in the supplemental information, to show the results of the spectral analysis.

→ The corresponding MTM figure has been added in the supplements.

Lines 349-352: this is confusing and difficult to follow as written. Can you add to Figure 7 the correlation lines between d18O and E+T-P?

→ We suspect that this is a misunderstanding. We have not used the isotopic curves for tuning. We only used them to position the interval that is to be tuned such that we can then tune the colour data to the correct E+T-P target. For this, the correlation lines are all shown. Also, the resulting effect on the isotopic curves is then shown in supplementary Figure S3. We understand where the misunderstanding arose and the sentence has been changed.

Line 374: "the CaCO3 AR is driven by both the carbonate content and the SR" I disagree with this statement. The correlation between CaCO3 AR and SR during KM5 has a R2=0.089. This means that there is no correlation between the two parameters.

→ Yes, but still significant according to the p-value (1.1 x 10^-2), therefore, we cannot reject the hypothesis, the sentence has been modified.

Figure 9, panel a: apart from the MCO, the panel shows that:

i) there is no correlation between CaCO3 AR and SR;
ii) at one single SR value corresponds different CaCO3 AR values. Can this result from the method used to build the age model? Or is there an oceanographic reason instead? I think the authors need to discuss this in the text. It seems to me that the fact that SR are linear plays a significant role in the relationship between CaCO3 AR and SR.

→ The presence or absence of correlation is tested by calculating the significance of the correlation coefficient. This reveals that the hypothesis for a higher-than-random correlation between SR and $CaCO_3$ AR can only be rejected for MIS5, but not for MIS9 and the Pliocene. We fully agree that the differences in the R value are enormous, and interpret the data accordingly, but we cannot ignore the results of the statistical test. The paragraph has been modified.

With respect to the second comment, we included a statement to the discussion of the results.

Figure 9, caption: regression lines of MIS 5 and MIS9 are difficult to distinguish. I suggest to add regression formula to the figure legend, to better represent the slope of regression lines.

→ The regression formula has been added on the figure.

R#1 : Discussion: I found the discussion a bit difficult to read and not well organized (see comments below). In addition, I couldn't find any discussion and interpretation of the new stable isotope records, which is a bit of shame considering that they can provide important information for the interpretation of the other records presented. In my opinion, a discussion on the stable isotope records and on how they correlate with sedimentation and accumulation rate records needs to be added.

R#2 : 4. I'm missing a section in the result part for the new δ18O, δ13C data generated from this study. Moreover, these data can provide additional information that can help the part of discussion.

→ We modified the discussion section around line 348 to add some comments on the stable isotope curves.

Paragraph 4.1: The discussion in this section is difficult to follow and needs rewriting.

→ Indeed, this section has been entirely restructured.

Lines 402-403: How can you reconcile your observation of dissolution in Pleistocene sediments with the fact that Site 927 has been located above the lysocline?

→ Above the lysocline TODAY (Cf point 1.b) from R#2 at the beginning of this document) but the Antarctic deep water formation and circulation during the glacial intervals is causing dissolution during these cold intervals (Curry et al., 1995; Frenz et al., 2006).

Lines 411-414: deleted this sentence.

→ This sentence has been reformulated in the new version.

Lines 424-425: I do not agree. Pelagic carbonate AR can indicate both carbonate production and carbonate dissolution. So how can carbonate production be assessed by carbonate SR without considering carbonate dissolution?

→ We modified the first sentence, to make our point clearer.

Figure 12: It is not clear what the dashed curves are. I suggest to change the names of curves in the legend.

→ The dashed curves are corresponding to the significance at 90% intervals. The names of curves in the legend have been changed.

Line 503: Please describe briefly the main observations made by the cited studies.

→ Yes, it has been done in the revised version of the manuscript.

Line 506: in my opinion it cannot be said that the new record has a similar long-term trend as Lyle et al. (2019).

→ A more appropriate description of the observations has been done in the revised version of the manuscript.

Conclusions: I suggest to shorten the conclusions which are extensively long.

→ We believe the main messages should be briefly summarised here and we also believe the conclusions are not longer than in other studies of this kind. We have considered this comment carefully and deleted is the sentence on line 520-522.

--------------- Technical Corrections --------------- Minor comments

Lines 20-21: "…, but that each interval was characterized by large orbital-scale variability" Although I understand what you mean, reword if possible.

→ This sentence has been reworded.

Lines 23-24: "…on orbital time scales similarly or even more than on longer time scales". Rephrase.

→ This sentence has been reworded.

Lines 43-44: quantify short-term and long-term.

→ "[...] geological (Ma) and orbital (ka) [...]" have been added in the text.

Line 71: "plankton had no opportunity to responds to the climate cycles by migration" Add a reference.

→ This sentence has been reworded.

Line 80: "… to assess the spatial coherence of long terms

→ The "s" at the end of "terms" has been deleted.

Line 111: "…is also characterised as wetter" wetter compared to today? clarify

→ This sentence has been clarified, specifying that indeed, it is compared to today.

R1 : Line 124: add colour scale for the bathymetry next to the map of Figure 1.
R2 : Figure 1: Add scale for bathymetry

→ Yes, a colour scale for the bathymetry has been added on this figure.

Line 130: "This aseismic ridge rises several km above…" Give depth

→ The depth has been added in the text.

Line 131: add "modern" before "regional". Add lysocline depth.

→ This has been modified in the revised version of the manuscript.

Line 161: What does "loess" mean in the plot vertical axis? If a detrending function was applied to the record, state it in the figure caption.

→ Yes, this is a smoothing method, and it has been added in the figure caption.

Line 164: substitute "Stable oxygen isotopes" with "Oxygen stable isotopes".

→ Yes, this has been modified in the revised version of the manuscript.

Line 168: delete "Next,".s

→ Yes, this has been modified in the revised version of the manuscript.

Line 193: substitute "S3" with "S1".

--> No, it is well the table S3 of the Westerhold et al. (2020) paper.

Line 200: substitute "For the high resolution 4 intervals" to "For the four high resolution intervals".

→ Yes, this has been modified in the revised version of the manuscript.

Figure 3: This is a nice figure. 3b: I recommend to add also a small key-scale showing the values of the colors.

→ Yes, a colour scale has been added on this figure.

Line 223: state that the graphs in panel a are box plots.

→ Yes, this has been modified in the revised version of the manuscript.

Line 253: the Leg number can be removed.

→ Yes, the Leg number has been removed in the revised version of the manuscript.

Lines 255-256: it is difficult to understand which color is which. I suggest to add a legend next to the panel.

→ Yes, this has been modified in the revised version of the manuscript.

Line 258: substitute "blue" with "light blue". Apply the same to figures 6 and 7.

→ Yes, this has been modified in the revised version of the manuscript.

Figure 6, panel f: I suggest to use another color instead of the light purple for the MS record of the middle core because it is difficult to distinguish from the MS record in dark purple.

→ Yes, this has been modified in the revised version of the manuscript.

Line 273: "… and times of fastest sea-level change..." What do you mean by fastest sea level change?

→ Specification on this have been added in the text.

Line 280: add corresponding colour for the MS record and the MS smoothed record.

→ Corresponding colour for the MS record and the MS smooth record have been added in the revised version of the manuscript in brackets (black and grey).

Line 298: add a brief explanation of why this insolation curve has been used.

→ A brief explanation has been added.

Line 213: which curve is obliquity and which is E+T-P?

→ Obliquity in grey, E+T-P in black, this has been specified in the legend.

Line 230: define "LAD".

→ Last appearance datum, this has been added in the revised version of the manuscript.

Line 358: substitute "local" with "Site 927".

→ This has been corrected in the revised version of the manuscript.

Line 368: substitute "periods" with "intervals".

→ This has been corrected in the revised version of the manuscript.

Line 381: "carbonate AR appears to decrease with time". Do the authors mean with increasing age?

→ Yes, with increasing age. This has been specified and this sentence has been reworded in order to make it clearer in the revised version of the manuscript.

Line 443: delete "On the other hand,"

→ This has been deleted in the revised version of the manuscript.

Line 506: "a largely similar overall trend…" I cannot see that - reword this part.

→ This has been reworded in the revised version of the manuscript.

Lines 507-508 you state "Clearly, the overall of carbonate accumulation at the Ceara Rise supports the existence of a late Miocene carbonate maximum also under tropical conditions". However, in lines 15-16 you note that there is "a systematic increase in sedimentation rates since the late Miocene, but carbonate accumulation rate does not show a clear trend", which is what your data show. Therefore, these lines in the discussion need rewording.

→ Indeed, what needed to be reworded was the claim that there is no trend in $CaCO_3$ AR at Ceara Rise (line 239), which we have now corrected, following a similar comment to this end by both referees.

Lines 515: "… The two shallow sites consistently…." Add sites in a parenthesis to remind them to the reader.

→ The site's references have been added in parenthesis in the revised version of the manuscript.

Figures 1, 3, 5, 6, 9, 12, S1 and S3 have been modified according to the reviewer's comments and the figures S4, S5, S6 and S7 have been added to the supplements.